# Association of Individual Health Literacy with Preventive Behaviours and Family Well-Being during COVID-19 Pandemic: Mediating Role of Family Information Sharing

**DOI:** 10.3390/ijerph17238838

**Published:** 2020-11-27

**Authors:** Janet Yuen Ha Wong, Abraham Ka Chung Wai, Shengzhi Zhao, Faustina Yip, Jung Jae Lee, Carlos King Ho Wong, Man Ping Wang, Tai Hing Lam

**Affiliations:** 1School of Nursing, Li Ka Shing Faculty of Medicine, The University of Hong Kong, Hong Kong, China; janetyh@hku.hk (J.Y.H.W.); lubabezz@connect.hku.hk (S.Z.); leejay@hku.hk (J.J.L.); 2Emergency Medicine Unit, Li Ka Shing Faculty of Medicine, The University of Hong Kong, Hong Kong, China; awai@hku.hk (A.K.C.W.); faustina@connect.hku.hk (F.Y.); 3Department of Family Medicine & Primary Care, Li Ka Shing Faculty of Medicine, The University of Hong Kong, Hong Kong, China; carlosho@hku.hk; 4School of Public Health, Li Ka Shing Faculty of Medicine, The University of Hong Kong, Hong Kong, China; hrmrlth@hku.hk

**Keywords:** COVID-19, health literacy, information sharing, family well-being, preventive measures

## Abstract

*Objective*: We tested a model of individual health literacy information sharing with family members, personal preventive behaviours and family well-being during the Coronavirus Disease 2019 (COVID-19) pandemic in Hong Kong. *Methods*: We analysed data of 1501 randomly selected Chinese adults from a cross-sectional survey in Hong Kong from 9 to 23 April, 2020. Individual health literacy about COVID-19 with the items extracted from the questionnaire in World Health Organization Risk Communication and Community Engagement (RCCE) Action Plan Guidance for COVID-19 preparedness and response, COVID-19 information sharing with family members, preventive behaviours against COVID-19 and family well-being were measured. Structural equation modelling analysis tested the proposed model. *Findings*: COVID-19 information sharing with family members partially mediated the association between individual health literacy and personal preventive behaviours. The direct effect of 0.24 was shown, and the indirect effect through COVID-19 information sharing with family members was small at 0.03 (Z = 3.66, *p* < 0.001). Family well-being was associated with personal preventive behaviours against COVID-19. The model was adjusted for sex, age, and socioeconomic status factors and had good fit with RMSEA = 0.04, CFI = 0.98, TLI = 0.96, and SRMR = 0.02. *Conclusion*: COVID-19 information sharing with family members was a partial mediator between individual health literacy and personal preventive behaviours against COVID-19. Strategies for enhancing health literacy and preventive measures against COVID-19 are needed to promote family well-being in the pandemic.

## 1. Introduction

To fight against the Coronavirus Disease 2019 (COVID-19) pandemic, the public’s health literacy and engagement in preventive measures, and vigilance about the disease and infection control, are essential [1]. Health literacy refers to the ability of people to access, understand and use information to make decisions related to their health [2]. It is an individual autonomous set of knowledge, and one with an adequate level of health literacy can take responsibility for one’s own and family health [3]. Health literacy, in the context of COVID-19, includes the knowledge and awareness of disease and preventive behaviours such as observing hand hygiene, the use of face masks, social distancing, traveling restrictions and sanitation.

Previous studies have shown the association between health literacy and health behaviours, including healthy lifestyles, vaccination and cervical cancer screening [4,5,6]. During the COVID-19 outbreak, personal preventive behaviours have been positive mitigation measures. As people have been instructed to stay at home to minimise the chances of getting infection or spreading to others, family support and resources have been particularly influential for enhancing health literacy and preventive behaviours against COVID-19. According to Bowen Family Systems Theory [7], family is a collection of people who are integrated and who interact with and are interdependent on each other. Each family system has its boundary that is selectively permeable to outside forces, including COVID-19 information, entering or leaving the system. Edwards and colleagues [8] defined sharing information among family members as part of a concept named “distributed health literacy”. They explored how health literacy is distributed around an individual by family members and found they shared knowledge and understanding, accessing and evaluating information, supporting communication and decision-making. Therefore, information sharing in families should contribute to personal preventive behaviours. However, we found some reports on COVID-19 information sharing on social media [9,10] but no reports on the COVID-19 information sharing among family members in enhancing personal preventive behaviours.

Bowen Family Systems Theory [7] also can add to our understanding on the coping of families during COVID-19 pandemic. The pandemic is a crisis that affects everyone, hence, the family system could face disequilibrium [11]. The imbalance might burden family members. During the transition, family well-being could be enhanced or decreased [11]. In the Chinese context, our previous qualitative study in Hong Kong showed that family well-being consisted mainly of family harmony, which leads to family health and happiness [12]. Additionally, our subsequent quantitative study found that core elements of family well-being were family health, harmony, and happiness [13]. We found no report on how the pandemic affected family well-being.

Hence, we investigated the aforementioned relations by testing a structured model. In this model, we firstly examined the mediating role of family information sharing about COVID-19 between individual health literacy and personal preventive behaviours. Then, we evaluated whether the examined mechanism with personal preventive behaviours was associated with family’s well-being [14] (Figure 1). The model was adjusted for sex, age and socioeconomic factors, which were the potential confounders.

## 2. Methods

### 2.1. Respondents and Setting

We used data from a population-based cross-sectional COVID-19 Health Information Survey (CoVHInS) conducted in Hong Kong from 9 to 23 April 2020, during the period of second wave of COVID-19 due to imported cases from Hong Kong returners; 1035 cases were confirmed and 4 deaths were confirmed by 23 April. At that moment, the Government tightened the anti-epidemic regulations, including, for example, closing bars, closing schools and implementing stringent travel restrictions. Hong Kong residents aged 18 or older and able to speak in Cantonese, Mandarin or English were eligible.

### 2.2. Measurements

#### 2.2.1. Health Literacy about COVID-19

Individual health literacy about COVID-19 was assessed by four constructs: (1) knowledge of COVID-19 (four items), (2) awareness of COVID-19 (two items), (3) knowledge about personal preventive behaviours to prevent COVID-19 (one item), and (4) knowledge about health seeking behaviours after infection (one item). The items were extracted from the questionnaire in World Health Organization (WHO) Risk Communication and Community Engagement (RCCE) Action Plan Guidance for COVID-19 preparedness and response [15]. A correct answer was given a score of one point, and any false or unknown answer was given a score of zero. The total health literacy score could range from 0 to 46, with higher scores indicating better knowledge about COVID-19. Its internal consistency was acceptable with Cronbach’s alpha 0.69.

#### 2.2.2. Personal Preventive Behaviours

We also assessed the number of personal preventive behaviours against COVID-19, including hand hygiene, before eating and after toileting, washing hands with alcohol-based sanitizer, wearing surgical masks or fabric masks when going out, adding water/bleach to household drainage system, keeping social distance from people in public areas (e.g., 1.5 m), reducing social contact with relatives/friends/neighbours and using alcohol/bleach to clean daily necessities. Participants were asked to respond by using a 4-likert scale (0 = never, 1 = seldom, 2 = sometimes and 3 = often). The total scores were the sum of all items range 0–27 (Appendix A). Its internal consistency was satisfactory with Cronbach’s alpha 0.72.

#### 2.2.3. COVID-19 Information Sharing with Family Members

One item assessed the frequency of sharing information about COVID-19 with family members in the past 3 months, by using a 4-likert scale (1 = never, 2 = occasionally, 3 = sometimes and 4 = often).

#### 2.2.4. Family Well-Being

Family well-being was assessed using family health, harmony and happiness (3H), which were shown to be the core elements of family well-being in Hong Kong [13]. The three questions were “Do you think your family is healthy/harmonious/happy?” with responses from 0 to 10 points. The higher the composite score, the better the family well-being. Its internal consistency was satisfactory with Cronbach’s alpha 0.89.

#### 2.2.5. Demographics and Socioeconomic Status

Information on demographic characteristics such as sex, age and socioeconomic status (SES), including education attainment, employment status, personal income, type of housing and receiving social security assistance indicating poverty, was collected.

### 2.3. Data Collection Procedures

Data were collected using telephone survey and online survey via text message invitations. The telephone interviews were conducted by experienced interviewers using a web-based Computer-Assisted Telephone Interview (CATI) system. A list of randomly generated landline telephone numbers with a geographically representative sample in Hong Kong was obtained via the system. All numbers were called up to five times before confirmation of unsuccessful status. To randomise the choice of the individual participants in each household, the interviewer asked to speak to a household member who would have a birthday nearest to the interview date. Informed consent was provided to the sampled residents prior to the interviews. Of 816 successful telephone number sampled, 500 eligible residents completed the telephone interviews (response rate 61.3%). On-site supervision and quality-control measures were implemented to ensure data quality.

To expedite the recruitment process, the online survey was conducted from a panel of residents’ database by sending text message invitations to randomly listed mobile phone numbers generated through the Numbering Plan for Telecommunication Services in Hong Kong provided by the Office of the Communications Authority (OFCA), with 100,790 residents covering diverse socio-economic backgrounds who agreed to be members of the panel. Before commencing the survey, text messages for invitation were sent to the selected panel members. After obtaining informed consent on an online platform, they were provided an online self-administered questionnaire. Using stratified random sampling by sex and age, 1623 invitation text messages were sent, and 1001 eligible residents completed the online questionnaire (response rate 61.7%).

### 2.4. Data Analysis

The data were weighted by sex, age and education to compensate for disparities between the sample and the general population aged 18 or older in Hong Kong by using data from Census and Statistics Department of Hong Kong. Descriptive statistics were used to describe the characteristics of sample and personal preventive behaviours. Pearson’s correlation coefficients were checked on all relevant variables. To test the hypotheses 1 to 2, Pearson’s correlation coefficients were checked on all relevant variables. Post hoc power analysis was conducted using correlation coefficients and reported satisfactory (power for analysis: 1.00). To test the proposed model, structural equation models (SEM) were performed to test the causal assumptions derived from the proposed model (Figure 1) by using Mplus Version 7.4 (Muthén and Muthén, Version 7.4, Los Angeles, CA, USA). It enables incorporation of measured and latent variables to be analysed in the same mode without problems of multi-collinearity. Specifically, in Model A, SEM examined the pathway between individual health literacy and personal preventive behaviours via COVID-19 information sharing with family members according to the procedures described in Preacher and Hayes [16]. In Model B, confounding variables, including sex, age and SES variables, were added into Model A to further assess the results. In Model C, we tested whether personal preventive behaviours were positively associated with family well-being as the outcome with the adjustment of sex, age and SES variables. As the normality of our sampling distribution of the total and specific indirect effects was unknown, bootstrapping of size 10,000 was used for estimating the indirect effects.

The adequacy of our hypothesized SEM models was evaluated by examining the overall goodness-of-fit of the models, including root mean squared error of approximation (RMSEA) being smaller than 0.06, the Comparative Fit Index (CFI) and Tucker–Lewis Index (TLI) larger than 0.95, and the standardized root mean squared residual (SRMR) smaller than 0.05 [17]. Moreover, Sobel test was used to assess the significance of indirect effects [16]. The direction and size of the indirect effects of the mediation analysis were also assessed. All statistical tests were two-sided. A *p*-value of less than 0.05 is considered statistically significant.

### 2.5. Ethical Approval

Ethical approval was obtained from the Institutional Review Board of the University of Hong Kong/Hospital Authority Hong Kong West Cluster (UW-20-238). We followed STROBE guidelines in reporting.

## 3. Results

### 3.1. Participant Characteristics

Table 1 shows that, of 1501 respondents, 47.8% were male and 27.8% aged 60 or above, 76.8% had secondary or tertiary education, 66.1% were married/cohabiting, 37.9% were living in public housing and 62.9% were currently employed. Post hoc power analysis was conducted using correlation coefficients and reported satisfactory (power for analysis: 1.00).

The mean score of health literacy on COVID-19 was 38.35 out of 56 (SD = 4.90). In general, the respondents had quite high score of personal preventive behaviours (mean = 20.30 out of 27, SD = 3.74). Washing hands before eating (74.5%) and after using the toilet (90.4%) and wearing a surgical mask when going out (82.9%) were the top three often performed behaviours (Table 2). Among all behaviours, wearing a fabric mask when going out (17.7%) was the least practiced, but wearing any mask was 89.3%.

COVID-19 information sharing with family members was associated with individual health literacy (Pearson’s r = 0.28, *p* < 0.0001) and associated with personal preventive behaviours (Pearson’s r = 0.21, *p* < 0.0001). Additionally, individual health literacy was associated with personal preventive behaviours (Pearson’s r = 0.29, *p* < 0.0001).

### 3.2. Model Testing

Table 3 shows the direct and indirect effects of the SEMs. In Model A, medium direct effect was 0.26 and a small indirect effect through COVID-19 information sharing with family members was shown as 0.04 (Z = 4.64, *p* < 0.001) between health literacy and preventive behaviours. The model had good fit with RMSEA = 0.00, CFI = 1.00, TLI = 1.00, SRMR = 0.00 and BIC = 26,653 (Figure 2). Model B, after adding sex, age and all SES variables, showed the medium direct effect of 0.26 and small indirect effect through COVID-19 information sharing with family members of 0.03 (Z = 3.90, *p* < 0.001) between health literacy and preventive behaviours. The model had marginally good fit with RMSEA = 0.06, CFI = 0.94, TLI = 0.78, SRMR = 0.02 and BIC = 19,885 (Figure 2).

Model C, after adding family well-being in Model B, showed essentially the same results as a medium direct effect of 0.24 and indirect effect through COVID-19 information sharing with family members was 0.03 (Z = 3.66, *p* < 0.001) between health literacy and preventive behaviours (Z = 9.53, *p* < 0.001). The model had good fit with RMSEA = 0.04, CFI = 0.98, TLI = 0.96, SRMR = 0.02 and BIC = 37,612.

Figure 3 shows the standardized regression coefficients (β) of the SEM of Model C. Older age (β = 0.22, *p* < 0.001), higher education level (β = 0.16, *p* < 0.001) and higher personal income (β = 0.10, *p* = 0.002) were associated with higher health literacy, while receiving social security assistance (β = −0.17, *p* < 0.001) was negatively associated with health literacy. Being female (β = 0.18, *p* < 0.001) and higher personal income (β = 0.09, *p* = 0.005) were associated with personal preventive behaviours, while both living in public housing (β = −0.05, *p* = 0.04) and receiving social security assistance (β = −0.10, *p* = 0.001) were negatively associated with personal preventive behaviours. Additionally, personal preventive behaviours were associated with family well-being as the outcome (β = 0.08, *p* = 0.001).

## 4. Discussion

We have first shown a model of associations of health literacy with preventive behaviours and family wellbeing. COVID-19 information sharing with family members partially mediated individual health literacy and personal preventive behaviours after adjusting for sex, age, and socioeconomic status and adding family well-being as the outcome of the model. The small indirect effect (0.03) was significant. In a meta-analysis, among all preventive behaviours, use of masks and physical distancing were found to be effective in preventing person-to-person transmission [18]. We identified that respondents had a high percentage of wearing any kind of mask (89.3%). Our results were similar to another population-based telephone survey in Hong Kong (*n* = 1005) in which 98.8% reported wearing face masks when leaving home in March 2020 [19]. Although use of masks was voluntary in Hong Kong at the time of data collection, the high percentage indicated that people in Hong Kong had high level of compliance with preventive behaviours. In addition, our model has provided the first evidence that sharing COVID-19 information with family members was associated with such preventive behaviours, and with positive family well-being, which was analysed as the outcome. The relationships remained with adjustment for sex, age and SES variables.

In our model, apart from sharing COVID-19 information with family members, we found that female and higher personal income were associated with personal preventive behaviours while living in public housing, which was government subsidized and low-cost, and receiving social security assistance was negatively associated with personal preventive behaviours. Two previous reports shared that people in low SES status were at disproportionately high risk of contracting COVID-19 due to fewer resources for proper precautions, physical distancing and the high prevalence of multiple chronic conditions such as hypertension and diabetes [20,21]. Although the pre-existing vulnerability within the families might increase susceptibility to the consequences of the pandemic, our model has indicated that individual health literacy and sharing information with family members could enhance preventive behaviours and family well-being. Strategies to enhance health literacy and information sharing in family could be partially helpful for underprivileged individuals and families during the COVID-19 pandemic.

### 4.1. Limitations and Suggestions for Future Studies

Our study had several limitations. First, the data were collected from a population-based survey although it only reached those who had landlines or internet access. Weighting was done so that the results would better reflect the population situation. However, the sampling strategy could be improved by using stratified sampling from the postal address from the office of population censuses. Second, we used a cross-sectional design and the SEM results, which could not confirm causation. Third, our measurements on health literacy about COVID-19 relied on the extracted items from the questionnaire in World Health Organization (WHO) Risk Communication and Community Engagement (RCCE) Action Plan Guidance for COVID-19 preparedness and response. Its psychometric properties were unclear, which might have created information bias on health literacy about COVID-19. However, at the time when we designed the study, this was the only available questionnaire on measuring health literacy about COVID-19. Lastly, although our study findings showed that information sharing with family members was associated with enhanced preventive behaviours during the pandemic, further qualitative studies are needed to examine the nature and content of family communication and how these could enhance preventive behaviours.

### 4.2. Implications to Practice

Traditionally, health literacy refers to individual-level abilities. However, our results can inform interventions on both individual health literacy and information sharing with family members in the pandemic, particularly low-income families; this is urgently needed in our fight against the pandemic. Therefore, in the future, tangible skills and resources for family-distributed health literacy should be considered in health education and promotion contexts. Additionally, as females showed more preventive behaviours, health literacy enhancement programmes should target women as many are the most significant carers in their families. Relevant family-based conversations should be strengthened for the women so that they can influence the male family members more efficiently. Reinforcing family communication and health information dissemination among family members after learning is recommended.

## 5. Conclusions

We first showed that COVID-19 information sharing with family members was a partial mediator between individual health literacy and personal preventive behaviours against COVID-19. The findings can contribute to devising strategies for promoting preventive measures against the COVID-19 pandemic and sustainable measures in public empowerment towards epidemic emergency.

## Figures and Tables

**Figure 1 ijerph-17-08838-f001:**
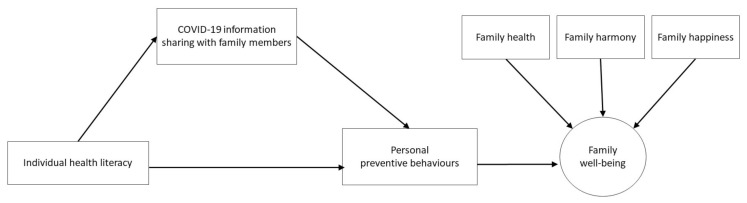
Proposed model of health literacy, preventive behaviours and family well-being.

**Figure 2 ijerph-17-08838-f002:**
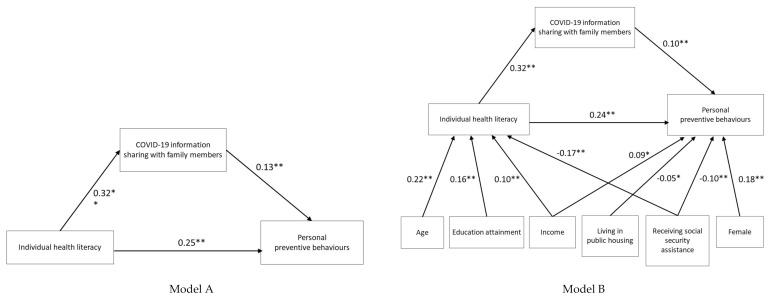
Standardized coefficients from the structural equations model (Model A, unadjusted and Model B, adjusted) for assessing the mediating effects of COVID-19 information sharing with family members on the relationships between individual health literacy and personal preventive behaviours. Note: The model was adjusted for sex, age and SES variables but only arrows with significant standardized coefficients are shown (* *p* < 0.05; ** *p* < 0.01).

**Figure 3 ijerph-17-08838-f003:**
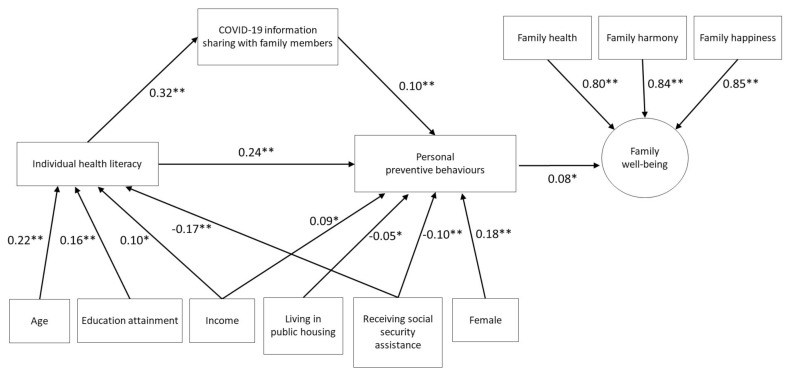
Standardized coefficients from the structural equations model (Model C) for assessing the mediating effects of COVID-19 information sharing with family members on preventive behaviours and family well-being.

**Table 1 ijerph-17-08838-t001:** Participant characteristics.

Characteristics	Sample Frame			Pooled (*n* = 1501)	
	Landline *n* (%) (*n* = 500)	Online *n* (%) (*n* = 1001)	Total *n* (*n* = 1501)	Unweighted (%)	Weighted (%) *
Gender					
Male	208 (41.6%)	464 (46.4%)	672	44.8%	47.4%
Female	292 (58.4%)	537 (53.6%)	829	55.2%	52.6%
Age (years)					
18–59	317 (63.4%)	689 (68.8%)	1006	67.1%	72.2%
60 or above	183 (36.6%)	312 (31.2%)	495	32.9%	27.8%
Marital status					
Married/cohabitated	355 (71.0%)	698 (69.7%)	1053	70.2%	66.1%
Unmarried/divorced/separated/widowed	145 (29%)	303 (30.3%)	448	29.9%	33.9%
Type of housing					
Public housing	204 (40.8%)	366 (36.6%)	570	38.0%	37.9%
Private housing	296 (59.2%)	635 (63.4%)	931	62.0%	62.1%
Employment status					
Unemployed/student/retiree/homemaker	179 (35.8%)	341 (34.1%)	520	34.7%	37%
Part-time work	46 (9.2%)	95 (9.5%)	141	9.4%	8.3%
Full-time work	275 (55.0%)	565 (56.4%)	840	56%	54.6%
Educational attainment					
Primary or above	88 (17.6%)	159 (15.9%)	247	16.5%	23.2%
Secondary	287 (57.4%)	577 (57.6%)	864	57.6%	45.4%
Tertiary or above	125 (25.0%)	265 (26.5%)	390	26.0%	31.4%
Monthly personal income (US$1 = HK$7.80)					
HK$ 10,000 or less	187 (37.4%)	332 (33.2%)	519	34.6%	37.5%
HK$ 10,001 ~ 20,000	161 (32.2%)	358 (35.8%)	519	34.6%	30.7%
HK$ 20,001 ~ 30,000	86 (17.2%)	182 (18.2%)	268	17.9%	17.5%
HK$ 30,001 ~ 40,000	34 (6.8%)	72 (7.2%)	106	7.1%	7.1%
HK$ 40,001 ~ 50,000	16 (3.2%)	29 (2.9%)	45	3.0%	3.5%
HK$ 50,001 or more	16 (3.2%)	28 (2.8%)	44	2.9%	3.8%
Receiving social security assistance	36 (7.2%)	62 (6.2%)	98	6.5%	10.3%

* Note: Weighted by sex, age and education to the general population in Hong Kong by using data from Census and Statistics Department of Hong Kong.

**Table 2 ijerph-17-08838-t002:** The number and weighted percentage of often performed personal preventive behaviours against COVID-19 (*n* = 1501).

Often Performed Personal Preventive Behaviours	*n* (%)
**Wash hands**	
Wash hands after using the toilet	1370 (90.4)
Wash hands before eating	1172 (74.5)
Wash hands with alcohol-based sanitisers	763 (49.4)
**Wear mask**	
Wear a surgical mask when going out	1258 (82.9)
Wear any kind of mask when going out	1357 (89.3)
**Other measures**	
Reduce social contact with relatives/friends/neighbours	635 (42.8)
Use alcohol/bleach to clean daily necessities	555 (36.0)
Add water/bleach to household drainage system	507 (34.1)
Keep distance from people in public areas	499 (33.8)
Wear a fabric mask when going out	251 (17.7)

**Table 3 ijerph-17-08838-t003:** The direct and indirect effect of individual health literacy on personal preventive behaviours through COVID-19 information sharing with family members.

Tested Models	Estimate	Product of Coefficients		*p*-Value	BC 95% CI
		SE	Z		
**Model A**					
Indirect	0.04	0.01	4.64	<0.001	(0.03, 0.06)
Direct	0.26	0.03	9.11	<0.001	(0.20, 0.31)
**Model B**					
Indirect	0.03	0.01	3.90	<0.001	(0.02, 0.05)
Direct	0.24	0.03	8.80	<0.001	(0.14, 0.23)
**Model C**					
Indirect	0.03	0.01	3.66	<0.001	(0.02, 0.05)
Direct	0.24	0.03	9.53	<0.001	(0.14, 0.23)

Note: SE = Standard Error; BC = Bias Corrected after Bootstrapping; CI = Confidence Interval.

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
