# Peer review of "Association of Individual Health Literacy with Preventive Behaviours and Family Well-Being during COVID-19 Pandemic: Mediating Role of Family Information Sharing"

_ijerph, 2020, doi:10.3390/ijerph17238838_

Round 1

Reviewer 1 Report

Thank you for this interesting and highly relevant paper about health literacy, preventive behaviours and family wellbeing during the novel corona virus pandemic. The article addresses important issues about health literacy and how this can be increased in the general population in times with the coronavirus. The paper is well-written and wellstructured, however I suggest some revision before publication, especially in the discussion section.

Background literature: The introduction suggests assocations between health literacy and health behaviour and that this might be influenced by information sharing in the family. The research field of health literacy has a separat concept of "distributed health literacy" e.g. sharing knowledge in families. Distributed health literacy includes  how people draw on their network for dealing with difficulties related to health. The introduction may benefit from including this concept for strenghtening the hypothesis.

Research aim: The research aim is divided in two parts. The first elements is supported by the introduction, however the second aim could be introduced better and it is not immediately clear what “coping process” means. Please be more clear about the argument for this aim

Methods: Health literacy is one of the key variables, but how did the authors choose the items from the WHO publication, and why did they not choose some of the existing HL scales?. How do the authors investigate the validation of the used questions - and in particulary the construct and criteria validity.. Discussion section: The authors should elaborate on the possible information bias due to the way they measured HL, and they should also discuss the possible selectionbias in the cohort investigated 

Author Response

Response to Reviewer 1 Comments

Re: Re-submission of Manuscript ID ijerph-959949
On behalf of my research team, I would like to thank you for taking your time and efforts in reviewing our manuscript. We have revised our manuscript in accordance with your comments and suggestions, and please find below our point by point responses. We hope that you will find our work suitable for publication in IJERPH.

Point 1: Thank you for this interesting and highly relevant paper about health literacy, preventive behaviours and family wellbeing during the novel corona virus pandemic. The article addresses important issues about health literacy and how this can be increased in the general population in times with the coronavirus. The paper is well-written and well structured, however I suggest some revision before publication, especially in the discussion section.

Response 1: Thank you so much for the positive comments from the reviewer. We have amended the manuscript according to your and other reviewers’ suggestions. We hope that you and the Reviewers will find our work suitable for publication in IJERPH.

Point 2: Background literature: The introduction suggests associations between health literacy and health behaviour and that this might be influenced by information sharing in the family. The research field of health literacy has a separate concept of "distributed health literacy" e.g. sharing knowledge in families. Distributed health literacy includes how people draw on their network for dealing with difficulties related to health. The introduction may benefit from including this concept for strengthening the hypothesis.

Response 2: We appreciate reviewer’s insightful suggestion. We have supplemented the concept of “distributed health literacy” in the introduction (page 1-2) and as follows:

“Edwards and colleagues defined sharing information among family members as part of a concept named as “distributed health literacy”. They explored how health literacy is distributed around an individual by family members and found they shared knowledge and understanding, accessing and evaluating information, supporting communication and decision-making. Therefore, information sharing in families should contribute to personal preventive behaviours.”

Reference: Edwards M, Wood F, Davies M, Edwards A. 'Distributed health literacy': longitudinal qualitative analysis of the roles of health literacy mediators and social networks of people living with a long-term health condition. Health Expectations 2015;18(5):1180-1193. doi:10.1111/hex.12093

Point 3: Research aim: The research aim is divided in two parts. The first elements is supported by the introduction, however the second aim could be introduced better and it is not immediately clear what “coping process” means. Please be more clear about the argument for this aim

Response 3: We agree with the comments. Coping process is something we have not yet explored in this study. We would like to clarify that the “coping process’ here means “ mechanism of the mediating role of family information sharing about COVID-19 information in the pathway between individual health literacy and personal preventive behaviours. We have made it clear in the introduction and as follows:

“In this model, we firstly examined the mediating role of family information sharing about COVID-19 between individual health literacy and personal preventive behaviours. Then, we evaluated whether the examined mechanism with personal preventive behaviours was associated with family’s well-being.”

Point 4: Methods: Health literacy is one of the key variables, but how did the authors choose the items from the WHO publication, and why did they not choose some of the existing HL scales?. How do the authors investigate the validation of the used questions - and in particulary the construct and criteria validity. Discussion section: The authors should elaborate on the possible information bias due to the way they measured HL, and they should also discuss the possible selection bias in the cohort investigated.

Response 4: We selected WHO publication for measuring HL about COVID-19 because there was no available validated tool to measure HL specifically to COVID-19.  We picked the items according to Knowledge-Attitude-Practice Model. Here are all the items:

Knowledge

  1. What do you know about Coronavirus Disease 2019 (or COVID-19)?
  • Wash hands before eating
  • Wash hands after using the toilet
  • Wear a surgical mask when going out
  • Wear a fabric mask when going out
  • Wash hands with alcohol-based sanitisers
  • Add water/ bleach to household drainage system
  • Keep distance from people in public areas (e.g. 1.5 metres)
  • Reduce social contact with relatives/ friends/ neighbours
  • Use alcohol/ bleach to clean daily necessities
  1. What are the main symptoms of the new coronavirus?
  • Fever
  • Cough
  • Shortness of breath and breathing difficulties
  • Muscle pain
  • Headache
  • Diarrhea
  1. How does the new coronavirus spread?
  • Blood transfusion
  • Droplets from infected people
  • Airborne
  • Direct contact with infected people
  • Touching contaminated objects/ surfaces
  • Sexual intercourse contact
  • Contact with contaminated animals
  • Mosquito bites
  • Eating contaminated food
  • Drinking unclean water
  1. Do you know how to prevent becoming sick from the new coronavirus?
  • Wear mask
  • Sleep under the mosquito net
  • Wash your hands regularly using hand rub or soap and water
  • Drink only treated water
  • Cover your mouth and nose when coughing or sneezing
  • Avoid close contact with anyone who has a fever and cough
  • Eliminate standing water
  • Cook meat and eggs well
  • Avoid unprotected direct contact with live animals and surfaces in contact with animals
  • Social distancing

Attitudes

  1. How dangerous do you think the new coronavirus is?
  2. I think that the coronavirus situation now is terrible.

Practices

  1. What have you and your family done to prevent becoming sick with coronavirus in the past 3 months?
  • Learn how to wear a mask appropriately
  • Washing hands regularly using alcohol-based cleanser or soap/ water
  • Only drink water that is filtered
  • Covering mouth and nose when coughing or sneezing
  • Avoid close contact with anyone who has a fever and cough
  • Eliminate standing water
  • Cook meat and eggs wells
  • Avoid unprotected direct contact with live animals and surfaces in contact with animals
  • Reduce social gathering
  1. What to do if you or someone from your family has symptoms of this disease?
  • Seek advice from others
  • Seek doctor’s diagnosis from a hospital
  • See a doctor at a clinic
  • See a Chinese medicine practitioner
  • Take prescription drugs
  • Self-quarantine at home

We have provided the questionnaire as supplementary document.

We agree with the reviewer that we shall discuss the possible bias. We have revised the section of “Limitations and suggestions for future studies” on page 12 and as follows:

“Third, our measurement on health literacy about COVID-19 rely on the extracted items from the questionnaire in World Health Organization (WHO) Risk Communication and Community Engagement (RCCE) Action Plan Guidance for COVID-19 preparedness and response. Its psychometric properties were unclear which might have information bias on Health literacy about COVID-19. However, at the time when we designed the study, this was the only available questionnaire on measuring health literacy about COVID-19.”

Reviewer 2 Report

Abstract

  1. You need to specify the instruments that were used to measure variables of interest in this study.

Materials and Methods

  1. Line 83-92: It is confusing that the total score of the 8-item health literacy scale ranges from 0-56. Please make sure you used the correct instrument and scoring system.
  2. Line 102-103: Please make sure the scoring system is correct. It was said a 5-Likert scale was used; however, the scoring system was not matched (i.e., 1-4).
  3. How did you perform informed consent for those who completed the on-line survey?
  4. What theoretical framework did you use to guide the proposed model and its casualty? The Family Systems Model does not map well onto your proposed model.
  5. What were the effect size and power of the study? How did you make sure your sample size was large enough to have sufficient power in this study?
  6. Please provide the figures to describe your three SEM models (e.g., A, B, C).
  7. Please report the reliability coefficients of all instruments used in this study.
  8. Please provide your survey questionnaire as an appendix.
  9. Please provide a bivariate correlation matrix of the variables measured in this study.

Results

  1. Please report the BIC values of each model.

Discussion

  1. Limitations of the study: There were ample limitations of study design that should be mentioned, such as sampling strategy, participant selection, etc. Please provide more information in this section.
  2. What is the uniqueness generated from your study? You have to discuss more information relevant to your research question and provide concrete suggestions to enrich existing knowledge of patient care.

Figure

  1. Figure 2
  • Please convert this figure in a table for better readability.
  • Please check the accuracy of the information in this table. “Wear a fabric mask when going out” was repeated in this figure.

Author Response

Response to Reviewer 2 Comments

Re: Re-submission of Manuscript ID ijerph-959949
On behalf of my research team, I would like to thank you for taking your time and efforts in reviewing our manuscript. We have revised our manuscript in accordance with your comments and suggestions, and please find below our point by point responses. We hope that you will find our work suitable for publication in IJERPH.

Abstract

Point 1: You need to specify the instruments that were used to measure variables of interest in this study.

Response 1: We have revised the abstract accordingly. 

Materials and Methods

Point 2: Line 83-92: It is confusing that the total score of the 8-item health literacy scale ranges from 0-56. Please make sure you used the correct instrument and scoring system.

Response 2: We apologize for the confusion. There was also a typo on the total score which should be ranged from 0-46. A supplementary document of the questionnaire was attached for showing the items of the measurement on health literacy about COVID-19. We selected WHO publication for measuring HL about COVID-19 because there was no available validated tool to measure HL specifically to COVID-19.  We picked the items according to Knowledge-Attitude-Practice Model. Here are all the items:

Knowledge

  1. What do you know about Coronavirus Disease 2019 (or COVID-19)?
  • Wash hands before eating
  • Wash hands after using the toilet
  • Wear a surgical mask when going out
  • Wear a fabric mask when going out
  • Wash hands with alcohol-based sanitisers
  • Add water/ bleach to household drainage system
  • Keep distance from people in public areas (e.g. 1.5 metres)
  • Reduce social contact with relatives/ friends/ neighbours
  • Use alcohol/ bleach to clean daily necessities
  1. What are the main symptoms of the new coronavirus?
  • Fever
  • Cough
  • Shortness of breath and breathing difficulties
  • Muscle pain
  • Headache
  • Diarrhea
  1. How does the new coronavirus spread?
  • Blood transfusion
  • Droplets from infected people
  • Airborne
  • Direct contact with infected people
  • Touching contaminated objects/ surfaces
  • Sexual intercourse contact
  • Contact with contaminated animals
  • Mosquito bites
  • Eating contaminated food
  • Drinking unclean water
  1. Do you know how to prevent becoming sick from the new coronavirus?
  • Wear mask
  • Sleep under the mosquito net
  • Wash your hands regularly using hand rub or soap and water
  • Drink only treated water
  • Cover your mouth and nose when coughing or sneezing
  • Avoid close contact with anyone who has a fever and cough
  • Eliminate standing water
  • Cook meat and eggs well
  • Avoid unprotected direct contact with live animals and surfaces in contact with animals
  • Social distancing

Attitudes

  1. How dangerous do you think the new coronavirus is?
  2. I think that the coronavirus situation now is terrible.

Practices

  1. What have you and your family done to prevent becoming sick with coronavirus in the past 3 months?
  • Learn how to wear a mask appropriately
  • Washing hands regularly using alcohol-based cleanser or soap/ water
  • Only drink water that is filtered
  • Covering mouth and nose when coughing or sneezing
  • Avoid close contact with anyone who has a fever and cough
  • Eliminate standing water
  • Cook meat and eggs wells
  • Avoid unprotected direct contact with live animals and surfaces in contact with animals
  • Reduce social gathering
  1. What to do if you or someone from your family has symptoms of this disease?
  • Seek advice from others
  • Seek doctor’s diagnosis from a hospital
  • See a doctor at a clinic
  • See a Chinese medicine practitioner
  • Take prescription drugs
  • Self-quarantine at home

Point 3: Line 102-103: Please make sure the scoring system is correct. It was said a 5-Likert scale was used; however, the scoring system was not matched (i.e., 1-4).

Response 3: We apologize for the typo. It should be 4-Likert scale. We amended accordingly.

Point 4: How did you perform informed consent for those who completed the on-line survey?

Response 4: Informed consent completion was conducted via online system. Firstly, the participants were provided an information sheet indicating the study information via the online survey platform. If participants had any question related to the study, they were encouraged to communicate via text message or phone call. Upon completion of the informed consent, the participants would be directed to the survey questionnaire. We revised the manuscript to make the information clear and as follows:

“After obtaining informed consent on an online platform, they were provided an online self-administered questionnaire.”

Point 5: What theoretical framework did you use to guide the proposed model and its casualty? The Family Systems Model does not map well onto your proposed model.

Response 5: We did not use any specific theory to guide the proposed model building. However, previous studies have already showed the strong association between health literacy and health behaviours. Family Systems Model has strengthened the proposed mediating role of COVID-19 information sharing with family members. In addition, we have supplemented the concept of “distributed health literacy” 

References:

  • Lorini, C., et al., Health literacy and vaccination: A systematic review. Human Vaccines & Immunotherapeutics, 2018. 14(2): p. 478-488.
  • Cajita, M.I., T.R. Cajita, and H.-R. Han, Health literacy and heart failure: a systematic review. The Journal of cardiovascular nursing, 2016. 31(2): p. 121.
  • Kim, K. and H.R. Han, Potential links between health literacy and cervical cancer screening behaviors: a systematic review. Psycho‐Oncology, 2016. 25(2): p. 122-130.

Point 6: What were the effect size and power of the study? How did you make sure your sample size was large enough to have sufficient power in this study?

Response 6: According to the previous study, prevalence of personal preventive behaviours in Hong Kong was above 60.0% (Cowling et al, 2020). A sample size of above 1000 is large enough in terms of identifying health literacy and preventive behaviours as compared with other cross-sectional studies during COVID-19 (Okan et al, 2020). By using the Exact test post hoc power analysis, the sample size of 1500 is of enough power for our main analyses, including Hypotheses 1 (individual health literacy and personal preventive behaviours, correlation coefficient 0.30, Power 1.00) and hypotheses 2 (information sharing with family members and personal preventive behaviours, correlation coefficient 0.22, Power 1.00).

Related texts have now been added in the Data analysis section as followed:

“To test the hypotheses 1 to 2, Pearson’s correlation coefficients were checked on all relevant variables. Post hoc power analysis was conducted using correlation coefficients and reported satisfactory (Power for analysis: 1.00).”

References:  

  • Cowling BJ, Ali ST, Ng TWY, et al. Impact assessment of non-pharmaceutical interventions against coronavirus disease 2019 and influenza in Hong Kong: an observational study. Lancet Public Health. 2020;5(5):e279-e288. doi:10.1016/S2468-2667(20)30090-6
  • Okan O, Bollweg TM, Berens E-M, Hurrelmann K, Bauer U, Schaeffer D. Coronavirus-Related Health Literacy: A Cross-Sectional Study in Adults during the COVID-19 Infodemic in Germany. Int J Environ Res Public Health. 2020;17(15):5503. doi:10.3390/ijerph17155503

Point 7: Please provide the figures to describe your three SEM models (e.g., A, B, C).

Response 7: We provided Model A and B as supplementary documents. For Model C, we would like to keep it as Figure 3 in the manuscript. 

Point 8: Please report the reliability coefficients of all instruments used in this study.

Response 8: We have 4 constructs in the proposed model. They were (1) individual health literacy, (2) COVID-19 information sharing with family members, (3) personal preventive behaviours and (4) family well-being. We have supplemented their respective reliability coefficients in the manuscript and as follows:

(1) Individual health literacy: “Its internal consistency was acceptable with Cronbach’s alpha 0.69.”

(2) COVID-19 information sharing with family members: This is assessed by single item. Therefore, no internal consistency can be calculated.

(3) Personal preventive behaviours:  “Its internal consistency was satisfactory with Cronbach’s alpha 0.72.”

(4) Family well-being: “Its internal consistency was satisfactory with Cronbach’s alpha 0.89.”

Point 9: Please provide your survey questionnaire as an appendix.

Response 9: We have provided the questionnaire as supplementary document.

Point 10: Please provide a bivariate correlation matrix of the variables measured in this study.

Response 10: Please find below for the bivariate correlation matrix of the variables. All of them are significantly correlated. 

Individual health literacy

COVID-19 information sharing with family members

Personal preventive behaviours

Family well-being: Healthy

Family well-being: Harmony

Family well-being: Happy

Individual health literacy

1

.29**

.30**

.07**

.06**

.07**

COVID-19 information sharing with family members

.29**

1

.21**

-.04**

-.05**

-.13**

Personal preventive behaviours

.30**

.21**

1

.11**

.04**

.10**

Family well-being: Healthy

.07**

-.04**

.11**

1

.65**

.67**

Family well-being: Harmony

.06**

-.05**

.04**

.65**

1

.69**

Family well-being: Happy

.07**

-.01**

.10**

.67**

.69**

1

Note: p-value <.01**

Results

Point 11: Please report the BIC values of each model.

Response 11: The BIC values for model A, B and C were 26653, 19885 and 37612 respectively. We have indicated in the manuscript.

Discussion

Point 12: Limitations of the study: There were ample limitations of study design that should be mentioned, such as sampling strategy, participant selection, etc. Please provide more information in this section.

Response 12: Thanks for the suggestion. We have provided more information about the “Limitations and suggestions for future studies” on page 12 and as follows:

“the data were collected from a population-based survey although it only reached those who had landlines or internet access. Weighting was done so that the results would better reflect the population situation. However, the sampling strategy could be improved by using stratified sampling from the postal address from the office of population censuses.”

“Third, our measurement on health literacy about COVID-19 rely on the extracted items from the questionnaire in World Health Organization (WHO) Risk Communication and Community Engagement (RCCE) Action Plan Guidance for COVID-19 preparedness and response. Its psychometric properties were unclear which might have information bias on Health literacy about COVID-19. However, at the time when we designed the study, this was the only available questionnaire on measuring health literacy about COVID-19.”

Point 13: What is the uniqueness generated from your study? You have to discuss more information relevant to your research question and provide concrete suggestions to enrich existing knowledge of patient care.

Response 13: As mentioned at the beginning of the Discussion section, our study is the first to show a model of associations of health literacy with preventive behaviours and family wellbeing. COVID-19 information sharing with family members partially mediated individual health literacy and personal preventive behaviours after adjusting for sex, age, and socioeconomic status and adding family well-being as the outcome of the model. Our findings can contribute to devising strategies, in particular about information sharing in families for promoting preventive measures against the COVID-19 pandemic and sustainable measures in public empowerment towards epidemic emergency.

Figure

Point 14: Figure 2

  • Please convert this figure in a table for better readability.
  • Please check the accuracy of the information in this table. “Wear a fabric mask when going out” was repeated in this figure.

Response 14: We revised accordingly.

Table 2. The number and weighted percentage of often performed personal preventive behaviours against COVID-19 (n=1501).

Often performed personal preventive behaviours

n (%)

Wash hands

Wash hands after using the toilet

1370 (90.4)

Wash hands before eating

1172 (74.5)

Wash hands with alcohol-based sanitisers

763 (49.4)

Wear mask

Wear a surgical mask when going out

1258 (82.9)

Wear any kind of mask when going out

1357 (89.3)

Other measures

Reduce social contact with relatives/ friends/ neighbours

635 (42.8)

Use alcohol/ bleach to clean daily necessities

555 (36.0)

Add water/ bleach to household drainage system

507 (34.1)

Keep distance from people in public areas

499 (33.8)

Wear a fabric mask when going out

251 (17.7)

Reviewer 3 Report

Public health messaging in the 21st century is complex with the widespread access to the internet and social media as well as the general decline in trust in authorities and institutions. The public and communities are affected by the infodemic that is accompanying the current pandemic. Therefore, the study of health literacy in such crisis conditions is of special importance, as a good understanding of health is a condition for following the preventive rules, and thus for creating resilient communities.
The study also presents the relationships between health literacy and family wellbeing, which has a relatively few literature, and even wellbeing has emerged only as an abstract target model rather than an operationalizable entity in health promotion practice.
The draft is well understood, concise, and the methodology is appropriate. The conclusions drawn from the results are modest.
At the same time, it should be noted that Hong Kong is a developed, relatively rich area of ​​China, it has already experienced disasters, such as the SARS epidemic in 2002, so the population is more sensitive and conscious to risk than in a rural environment. On the other hand, the correlations with regard to societal wellbeing are worth further examination. However, the manuscript is recommended for publication.

Author Response

Response to Reviewer 3 Comments

Re: Re-submission of Manuscript ID ijerph-959949
On behalf of my research team, I would like to thank you for taking your time and efforts in reviewing our manuscript. We have revised our manuscript in accordance with you and other reviewers’ comments and suggestions. We hope that you will find our work suitable for publication in IJERPH.

Reviewer: Public health messaging in the 21st century is complex with the widespread access to the internet and social media as well as the general decline in trust in authorities and institutions. The public and communities are affected by the infodemic that is accompanying the current pandemic. Therefore, the study of health literacy in such crisis conditions is of special importance, as a good understanding of health is a condition for following the preventive rules, and thus for creating resilient communities.
The study also presents the relationships between health literacy and family wellbeing, which has a relatively few literature, and even wellbeing has emerged only as an abstract target model rather than an operationalizable entity in health promotion practice.
The draft is well understood, concise, and the methodology is appropriate. The conclusions drawn from the results are modest.
At the same time, it should be noted that Hong Kong is a developed, relatively rich area of ​​China, it has already experienced disasters, such as the SARS epidemic in 2002, so the population is more sensitive and conscious to risk than in a rural environment. On the other hand, the correlations with regard to societal wellbeing are worth further examination. However, the manuscript is recommended for publication.

Response: We appreciate the positive feedback. We believe our findings can contribute to devising strategies, in particular about information sharing in families for promoting preventive measures against the COVID-19 pandemic and sustainable measures in public empowerment towards epidemic emergency.

Round 2

Reviewer 2 Report

Abstract

  1. Line 26: …behaviours and family well-being during COVID-19…

→ …behaviours and family well-being during the Coronavirus Disease 2019 (COVID-19)…

  1. Line 29: … extracted from the questionnaire in World Health Organization (WHO)…

→… extracted from the questionnaire in World Health Organization…

Materials and Methods

  1. Line 103: It is confusing that the total score of the 8-item health literacy scale ranges from 0-46. As you mentioned, “a correct answer was given a score of one point, and any false or unknown answer was given a score of zero.” Should the score range from 0-8? You may describe the scale encompasses four constructs by using XX items to measure health literacy. Please make sure you used the correct instrument and scoring system.
  2. Line 107: How many items did you use to measure personal preventive behaviors? Please provide the questionnaire as an appendix.

Discussion

  1. Line 270: …were unclear which might have information bias on Health literacy…

→ …were unclear which might have information bias on health literacy…

  1. What is the uniqueness generated from your study? You have to discuss more information relevant to your research question and provide concrete suggestions to enrich existing knowledge of patient care.

Author Response

Response to Reviewer 2 Comments (Round 2)

Abstract

Point 1: Line 26: …behaviours and family well-being during COVID-19…

→ …behaviours and family well-being during the Coronavirus Disease 2019 (COVID-19)…

Response 1: Thank you. We have amended accordingly.

Point 2: Line 29: … extracted from the questionnaire in World Health Organization (WHO)…

→… extracted from the questionnaire in World Health Organization…

Response 2: Thank you. We have amended accordingly.

Materials and Methods

Point 3: Line 103: It is confusing that the total score of the 8-item health literacy scale ranges from 0-46. As you mentioned, “a correct answer was given a score of one point, and any false or unknown answer was given a score of zero.” Should the score range from 0-8? You may describe the scale encompasses four constructs by using XX items to measure health literacy. Please make sure you used the correct instrument and scoring system.

Response 3: Thank you for reviewer’s reminder. To clarify, please find the below items for measuring the health literacy.

Health literacy on COVID-19 based on knowledge, attitudes and practices

Knowledge

1.      What do you know about Coronavirus Disease 2019 (or COVID-19)?

(I don’t know anything about it/ A virus that can cause disease / A government’s scheme/ A TV/ radio event)

Explanation: “Coronavirus Disease 2019” refers to the cluster of viral pneumonia cases occurring in Wuhan, Hubei Province, since December 2019. (https://www.chp.gov.hk/en/healthtopics/content/24/102466.html)

2.      What are the main symptoms of the new coronavirus?

-        Fever (Y/N)     

-        Cough (Y/N)    

-        Shortness of breath and breathing difficulties (Y/N)       

-        Muscle pain (Y/N)       

-        Headache (Y/N)           

-        Diarrhea (Y/N)             

3.      How does the new coronavirus spread?

-        Blood transfusion (Y/N)

-        Droplets from infected people (Y/N)     

-        Airborne (Y/N)

-        Direct contact with infected people (Y/N)          

-        Touching contaminated objects/ surfaces (Y/N) 

-        Sexual intercourse contact (Y/N)           

-        Contact with contaminated animals (Y/N)          

-        Mosquito bites (Y/N)    

-        Eating contaminated food (Y/N)

-        Drinking unclean water (Y/N)   

4.      Do you know how to prevent becoming sick from the new coronavirus?

-        Wear mask

-        Sleep under the mosquito net

-        Wash your hands regularly using hand rub or soap and water      

-        Drink only treated water

-        Cover your mouth and nose when coughing or sneezing

-        Avoid close contact with anyone who has a fever and cough

-        Eliminate standing water

-        Cook meat and eggs well          

-        Avoid unprotected direct contact with live animals and surfaces in contact with animals

-        Social distancing

Attitudes

5.      How dangerous do you think the new coronavirus is?

(Very dangerous (i.e. life threatening)/ Dangerous (i.e. require hospitalisation)/ Somewhat dangerous (i.e. require to rest at home)/ Not dangerous (i.e. can maintain daily activities))

6.      I think that the coronavirus situation now is terrible.

(Strongly disagree/ Disagree/ Neutral/ Agree/ Strongly agree)

Practices

7.      What have you and your family done to prevent becoming sick with coronavirus in the past 3 months?

-        Learn how to wear a mask appropriately

-        Washing hands regularly using alcohol-based cleanser or soap/ water     

-        Only drink water that is filtered

-        Covering mouth and nose when coughing or sneezing

-        Avoid close contact with anyone who has a fever and cough       

-        Eliminate standing water

-        Cook meat and eggs wells

-        Avoid unprotected direct contact with live animals and surfaces in contact with animals

-        Reduce social gatherings

8.      What to do if you or someone from your family has symptoms of this disease?

-        Seek advice from others  

-        Seek doctor’s diagnosis from a hospital 

-        See a doctor at a clinic (Western medical doctors)

-        See a Chinese medicine practitioner

-        Take prescription drugs

-        Self-quarantine (at home)

Point 4: Line 107: How many items did you use to measure personal preventive behaviors? Please provide the questionnaire as an appendix.

Response 4: We have put the questionnaire for measuring personal preventive behaviours in the appendix.

Discussion

Point 5: Line 270: …were unclear which might have information bias on Health literacy…

→ …were unclear which might have information bias on health literacy…

Response 5: Thank you. We have amended accordingly.

Point 6: What is the uniqueness generated from your study? You have to discuss more information relevant to your research question and provide concrete suggestions to enrich existing knowledge of patient care.

Response 6: This is the first study to test a model of associations of health literacy with preventive behaviours and family wellbeing. The findings have informed future practice in health literacy intervention. We provided more concrete suggestions in the manuscript (page 12-13) and as follows:

“Implications to practice

Traditionally, health literacy refers to individual-level abilities. However, our results can inform interventions on both individual health literacy and information sharing with family members in the pandemic, particularly, for families with low income are urgently needed in our fight against the pandemic. Therefore, in the future, tangible skills and resources for family distributed health literacy should be considered in health education and promotion context. Also, as females showed more preventive behaviours, health literacy enhancement programmes should target women as many are the most significant carers in the families. Relevant family-based conversations should be strengthened for the women so that they could probably influence the male family members more efficiently. Reinforcing family communication and health information dissemination among family members after learning are recommended.”

Round 3

Reviewer 2 Report

This revised version improves its readability. Thanks!